# Interpreting the Trends of Extreme Precipitation in Florida through Pressure Change

Chi Zhang [1,*] , Songzi Wu [1], Tiantian Li [2], Ziwen Yu [1] and Jiang Bian [3]

1   Department of Agriculture and Biological Engineering, University of Florida, Gainesville, FL 32603, USA; wusongzi@ufl.edu (S.W.); ziwen.yu@ufl.edu (Z.Y.)
2   Department of Geosciences, Florida Atlantic University, Boca Raton, FL 33431, USA; lit@fau.edu
3   Department of Health Outcomes & Biomedical Informatics, University of Florida, Gainesville, FL 32611, USA; bianjiang@ufl.edu
*   Correspondence: chi.zhang1@ufl.edu

**Abstract:** Precipitation is one of the many important natural factors impacting agriculture and natural resource management. Although statistics have been applied to investigate the non-stationary trend and the unpredictable variances of precipitation under climate change, existing methods usually lack a sound physical basis that can be generally applied in any location and at any time for future extrapolation, especially in tropical areas. Physically, the formation of precipitation is a result of ascending air which reduces air pressure and condenses moisture into drops, either by irregular terrain or atmospheric phenomena (e.g., via frontal lifting). Thus, in this paper, pressure change events (PCEs) will be used as a physical indicator of the stability of atmospheric systems to reveal the impact of temperature on precipitation in the tropical areas of Florida. By using data from both national and regional weather observation networks, this study segments the continuous observation series into PCE sequences for further analysis divided by dry and wet seasons. The results reveal that the frequency and intensity of PCE are highly associated with the occurrences of weather events. Decreasing pressure favors precipitation, and may turn extreme when the temperature and air moisture are sufficient to fuel the process. With similar intensity, decreasing pressure change events (DePCEs) generally bear a higher probability of precipitation (POP) and precipitation depth (PD) than increasing pressure change events (InPCEs). The frequency of alternating between InPCEs and DePCEs is subject to the temperature of the season and climate. Due to the seasonal fluctuations of weather characteristics, such as temperature and relative humidity, the dependence of extreme precipitation on these characteristics can be interpreted via PCE. A 7% increase rate of precipitation vs. temperature rise, determined by the Clausius—Clapeyron (C—C) relationship, can be observed from extreme precipitation with variances in the season and PCE types. Although indicated by other research, active vertical movement of air caused by a phase change in water at the frozen point is not pronounced in Florida. The response patterns of humidity to precipitation also vary by season and PCE types in extreme conditions. In summary, PCEs demonstrate reliable physical evidence of precipitation formation and can better associate the occurrence and intensity of extreme weather with other characteristics. In turn, such associations embody the underlying physical concepts present at any location in the world.

**Keywords:** precipitation; pressure change; weather extremes; Florida

## 1. Introduction

Trends in extreme weather events have changed drastically over recent decades [1], and there is plenty of evidence to support that the patterns of temperature and precipitation have changed with global warming. Climate change has led to an increase in the frequency and intensity of extreme weather events [2] such as heavy rain, drought, and heat waves, having a wide impact on production and economy, especially in agriculture.

For example, from 1964 to 2007, a crop loss of 1820 million and 1190 million Mg was caused by droughts (approximately equal to the global maize and wheat production in 2013) [3] and extreme heat disasters, respectively [4]. According to the most complete disaster reports for 2000–2007 [3], 6.2% of the total global cereal production was lost due to extreme weather events when compared to the expected total production. Florida is highly vulnerable to extreme weather, such as summer thunderstorms, hurricanes, drought, etc., bringing significant risks to the economic, environmental, and social systems. For example, Beaver et al. [5] mentioned that hurricanes in Florida could destroy submerged aquatic vegetation, which may break the balance of the ecosystem.

In order to model extremes for agriculture or hydrology projects, weather-related research is mainly statistically based, and assumes the dependence of extremes on climate conditions [6–10], which is restricted by the time and location to which the data are attributed, lacking portability. Therefore, finding a physically sound method to interpret the extreme weather events is important in forming a generality that applies regardless of the specifics of a dataset, which is usually an incomplete representation of all scenarios.

Physically, the formation of precipitation is a result of ascending air which reduces air pressure and condenses the moisture into drops, either by irregular terrain or by atmospheric phenomena (e.g., via frontal lifting) [11,12]. As justified by Yu et al. [13] in terms of the analysis of pressure change events (PCEs), precipitation is favored by decreasing pressure and amplified by rising temperatures, consistent with the Clausius–Clapeyron (C–C) relationship [14–19] under extreme conditions. Although the moisture-holding capacity will be high under high temperature, PCE determines the capability of extracting moisture into precipitation. However, not many studies have been conducted to investigate the interaction of temperature, pressure change and absolute humidity (AH) during extreme precipitation in PCE-determined weather events in tropical areas.

In this study, PCE will be used as a physical indicator for the stability of atmospheric systems in the development of extreme precipitation features subject to other weather measurements in the tropical area of Florida. There are three key components in this research: (1) compiling data from both national (NOAA) [20] and regional weather observation networks (FAWN) [21] and segmenting the continuous observation series into PCE sequences for further analysis; (2) defining the features of extremes (a detailed definition of extreme follows at the end of Section 2.1) from data, and extracting the dependence of different extremes on long-term temperature climate conditions (e.g., average monthly temperature (AMT)), in order to interpret the climate impact on extreme occurrences and intensity; And (3) refining and transforming the features that define the weather conditions associated with PCE, as well as investigating the dependence of extremes on these weather conditions.

This paper aims to justify, from agricultural and hydrology perspectives, the physical nexus of weather formation according to evidence gathered from analyzing various weather observations on a PCE basis, with an emphasis on extreme precipitation. In the rest of this paper, data sources and conditions will be introduced first, followed by the methodology section, which describes the pre-processing of data and how PCE is used to interpret the physical mechanism of precipitation formation. Next, the results section graphically inspects the relationships between PD/POP and cumulative event pressure change (CEPC), before investigating the dependence of AMT and atmospheric stability, in order to interpret the relationship between PCE and the extremes of precipitation in different seasonal weather changes. Summaries are given at the end.

## 2. Data

The analysis in this paper focuses on the state of Florida in the United States, a region characterized by humid tropical climate patterns in the south and a subtropical climate pattern (as shown in Figure 1) in the center and north [22]. Hourly observations of precipitation (mm), sea level pressure (hPa), air temperature (°C), and relative humidity (RH) (%) are used for analysis. To achieve good coverage of the overall state, data were collected from two sources: National Centers for Environmental Information (NCEI) [20]

for six main cities (see Figure 1) (Pensacola (ID:13,899: 1948–2021), Tallahassee (ID:93,805: 1942–2021), Jacksonville (ID:13,889: 1948–2021), Orlando (ID:12,815: 1952–2021), Tampa (ID:12,842: 1940–2021) and Miami (ID:12,839: 1948–2021)), and the Florida Automated Weather Network (FAWN) [21] for rural agriculture settings. FAWN is a weather network of 42 monitoring stations across Florida, and is maintained by the University of Florida. Figure 1 shows all of the locations. The data from all stations of both sources are lumped for analysis, since the physics of the dependence of extremes on weather parameters holds generally.

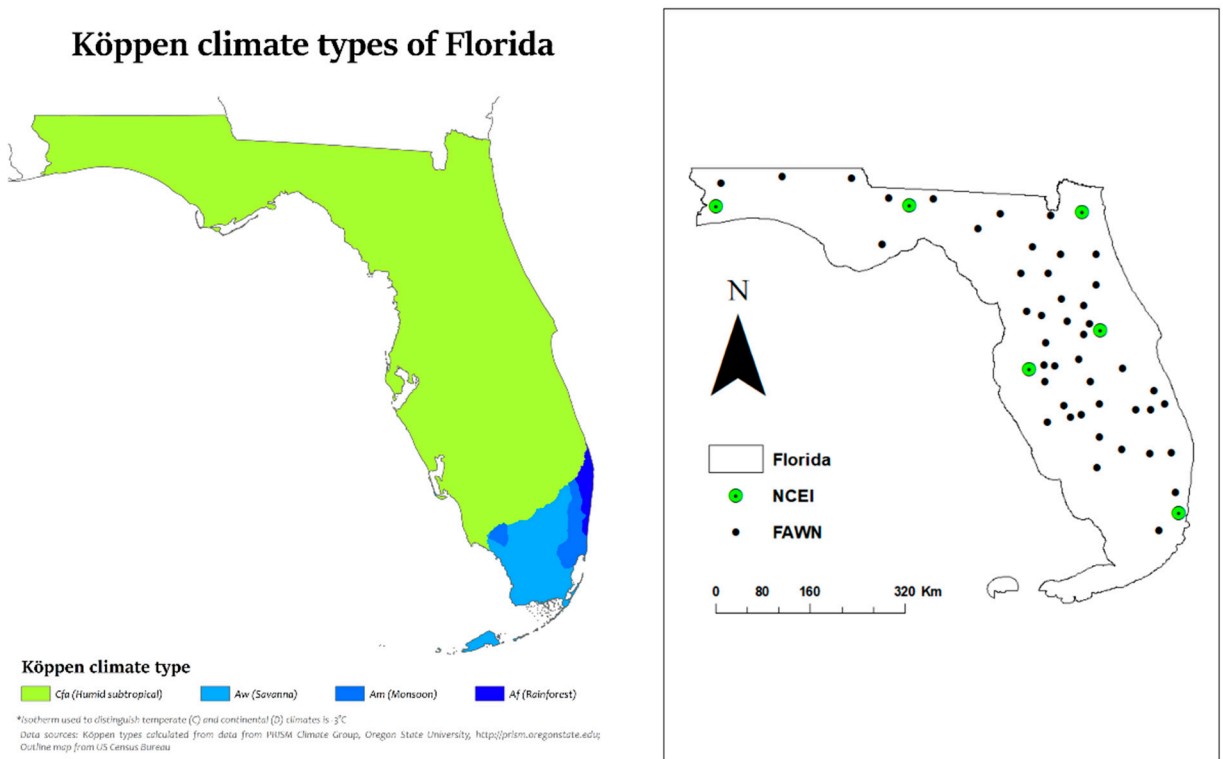

**Figure 1.** Positions of all weather observation locations in this study.

### 2.1. Data Preparation

There were multiple pre-processes performed before analysis. To increase the sample size of extremes in analysis, data from those two sources were converted into a consistent format and lumped in this study. Data quality was managed by processes of regulating time intervals, removing significant gaps, and interpolating remaining gaps. While data from NCEI are hourly, FAWN's data are in 15 min intervals and are aggregated into hourly intervals by summing up precipitation and averaging other measurements. From the raw data, 0.89% of all time intervals (2.00% of NCEI data and 0.10% of FAWN data) are greater than 1 h. Among these gaps, 0.03% are over 4 h and are considered too significant to be included in the analysis. For the remaining gaps (2 h~4 h) with a portion of 0.86% in the pooled data, the occurrences of the gap are summarized by each location and different gap lengths. Where a gap of a specific length frequently appears to be more than 100 times in a location, its impact will be considered unignorable and these gaps will be excluded.

To fill the remaining gaps, the method from Yu et al. [13] is adopted. The cubic (or Hermite) spline method [23] is used to interpolate the missing points in gaps less than 24 h. Then, a moving average method with a 24 h window is used to smooth the interpolated data. For the trivial gaps (between 6 and 24 h), data of 7 days centered on the day of interest are used to fit the spline method and interpolate these gaps as dry periods [5]. Therefore, all gaps will be filled by the spline method, which simulates the moving patterns of the neighbor points and connects the two ends of the gaps smoothly [24].

After processing, all the time series of the weather observations will be split into a sequence of PCEs [13] (see Section 3), the conceptual basis of this study, and a sequence of precipitation events. PCEs with any temporal overlap with the significant gaps identified at the beginning are removed from further analysis. In this study, an extreme climate condition is defined as a PCE with a magnitude of CEPC (positive or negative) surpassing 300 hPa (statistically, these extremes are less than 5% on both tails of the distribution of all CEPCs).

## 3. Methodologies

### 3.1. Character Definitions

Air mass movement is the principle of forming precipitation, leading to pressure change. Thus, the analysis should start by exploring the relationship between pressure changes and precipitation. In this study, pressure change and precipitation are both processed on an event basis. The precipitation event is determined by an inter-event dry period (IntEDP) [24], which has different quantile thresholds in order to avoid the influence of extreme diminutive events [17,25]. This paper uses 4 h for all cities as the minimum IntEDP, which is calculated by taking 99.5% as the quantile threshold. PCE is determined by the deseasonalized air pressure [13], which is the air pressure change lagged by 24 h (See Equation (1)).

$$P'(t) = P(t) - P(t-24) \tag{1}$$

$P'(t)$ is the deseasonalized air pressure at hour $t$, and $P(t)$ is the actual air pressure at hour $t$. This process helps remove the daily fluctuation of air pressure.

There are two types of PCEs: increasing pressure change events (InPCEs) and decreasing pressure change events (DePCEs) (see Figure 2), defined by the direction of pressure change. Using the data described in Section 2, the CEPC of InPCE ranges from 0 to 1100 hPa, and the CEPC of DePCE ranges from −1200 to 0 hPa. CEPC is defined as the cumulative air pressure change within a PCE (the shadowed area in Figure 2).

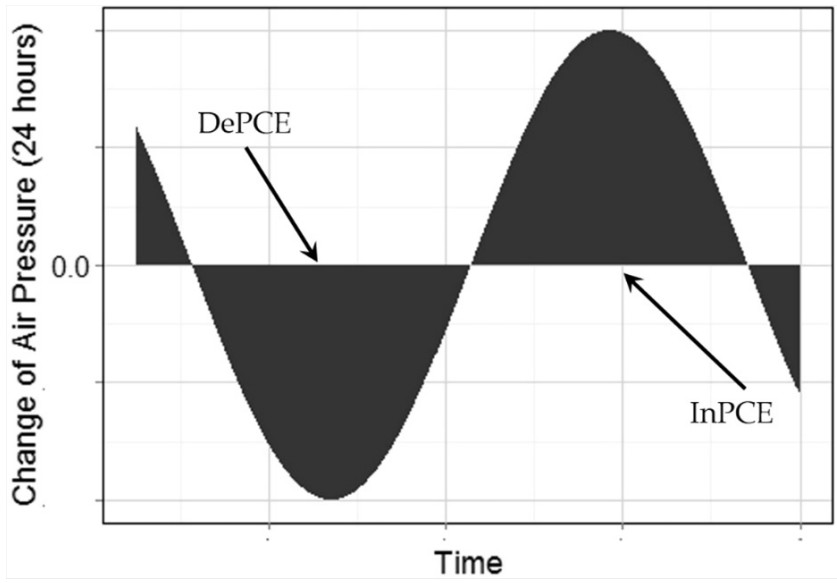

**Figure 2.** The definition of PCE and its types [13].

Next, the relationship between air pressure and hourly precipitation within a PCE is investigated. This analysis is used to study whether the historical hourly precipitation is related to air pressure on an event basis, by plotting the POP and PD over CEPC. POP is

referred to as the fraction of rainy PCEs in all PCEs (see Equation (2)). A rainy PCE is a PCE with precipitation.

$$POP = \frac{Num\_rainy\_PCE}{Num\_PCE} \tag{2}$$

where *Num_rainy_PCE* is the number of total PCEs having precipitation, and *Num_PCE* is the number of all PCEs. This calculation is performed by binning CEPC with 20 hPa intervals. For example, when CEPC ranges from $-480$ hPa to $-500$ hPa, there are 10 PCEs in total and 8 of them are rainy. Then, the POP is 8/10 = 0.8.

The calculation of AH follows the process given by Murray et al. [26], which can be separated into four steps: (1) converting the Fahrenheit temperature to Kelvin temperature; (2) depending on the temperature, calculating the saturation vapor pressure using the Murray equation; (3) calculating the partial water vapor pressure based on the relative humidity and saturation water vapor pressure at the same temperature; And (4) calculating AH based on partial water pressure and temperature.

*3.2. Analysis*

It is hypothesized that the stability of the atmospheric system (occurrence of PCEs) is highly correlated to the formation of weather events, including extremes. To verify this assumption, first, the frequency of alternating PCEs is graphically interpreted over AMT under different seasons. Then, the impact of PCE types on PD, POP, and seasonality is investigated by examining the relationship between different PD percentiles and AMT under the different scales of CEPC and seasons. The pattern of AH response to precipitation is also analyzed.

**4. Results and Discussion**

Figure 3 shows the relationship between PD/POP and CEPC. To characterize the correlation between POP and CEPC, a dashed local fit line via the loess method [13] is used. It is noteworthy that only the distribution of the rainy PCEs is represented by the density plot in the lower chart, while all the dry PCEs align with the *x*-axis where PD = 0 mm. The vertical line, where CEPC = 0 hPa, divides the field of POP into two distinct PCE types. Generally, the POP of DePCE (left region) increases with the absolute value of a CEPC, while the POP of InPCE rises at a slower rate and with a lower peak. The PCE reaches the highest frequency where CEPC = $-100$ hPa and PD = 20 mm (center of the contour). The trend of PD over CEPC is comparable with POP, which can be separated by InPCE and DePCE. The PD increases over CEPC magnitude in DePCEs but decreases in InPCEs. Based on the results above, we found that the POP of InPCE peaks at 40% when CEPC increases from 0 hPa to 100 hPa, and then decreases to 20% when CEPC reaches 480 hPa. For DePCE, the POP monotonously increases from 20% to 90% within 0~$-400$ hPa. Due to the limited sample size of extreme CEPC for both types, fluctuations are present at the tails on both ends. However, based on the results from a previous study focusing on NE US [13], DePCE eventually reaches a POP of 100% as CEPC grows beyond $-480$ hPa. Similarly, InPCE POP also flattens out in high CEPC, but at a POP lower than 100%. The association between POP, PD, and CEPC is prominent, and the falling air pressure appears to be more important than the increasing pressure in precipitation formation. Given that the atmospheric system in the subtropical area is relatively more stable than in the temperate zone and frigid zone, the CEPCs in this study are mostly distributed within the range between $-500$ hPa and 500 hPa. The extreme DePCE EPCs from $-800$ to $-600$ hPa represent a solid indication of precipitation, corresponding to a drastic POP increase. Physically, extreme DePCE occurs along with intensive convections, either locally or systematically, which favor the formation of precipitation. Because the ground observations were collected by point source stations, the sample size of local convections (e.g., daily thunderstorms in tropical areas) may be underestimated. Likewise, InPCE, which usually presents to balance a DePCE at the same time, may receive a marginal portion of PD. In turn, the chance of recording this portion

is subject to both the direction of precipitation movement and the location of the station. Therefore, the POP of InPCE is much lower, even for extreme CEPCs.

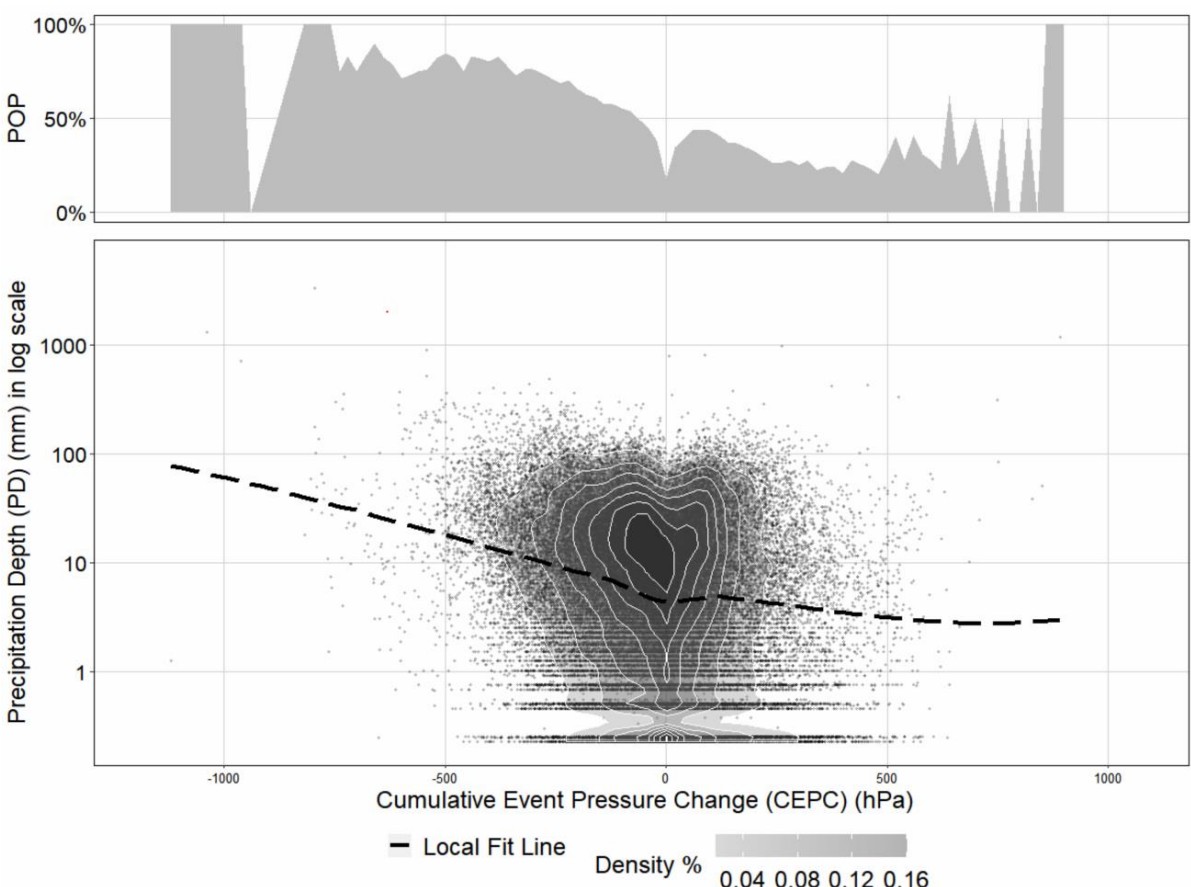

**Figure 3.** Relationship between precipitation depth (PD) and probability of precipitation (POP) vs. cumulative event pressure change (CEPC).

Bjerknes et al. [25] and Defant [27] suggest that the occurrence of air circulation and its corresponding air pressure change could be treated as an indicator of atmospheric stability, especially for moderate and intensive events. Since precipitation is formed due to atmospheric instability, it is important to evaluate the impact of temperature on the atmospheric system. Figure 4 illustrates the monthly frequency of moderate and intensive PCEs (|CEPC| > 90 hPa) against AMT in two half-years. Figure 4 is used to investigate whether PCEs associate with AMT. AMT is the average monthly temperature. The points in the plots represent the monthly aggregation of occurrence of PCE. A local regression line is added to highlight the trend. The patterns are similar in two half-years in that the monthly frequency goes down with the increase in AMT. Note that PCE in high temperature could be more intensive than in low temperature, even though the frequency is low. Based on the facts derived from Figure 4, there are several findings worth discussing. (1) The frequency of moderate and intensive PCEs over AMT is depicted graphically in Figure 4. A decreasing trend in the frequency of PCEs can be observed with an AMT increase. PCEs occur more frequently in low AMT, which means that an unstable atmospheric system mainly occurs when the temperature is relatively low. (2) It should be noticed that the PCEs in low AMT are relatively short and gentle compared with those in high AMT. Since atmospheric instability is attributed to low AMT, precipitation formation is favored and has a high chance of occurrence. However, the corresponding PD is relatively low due to the short and gentle PCEs. (3) On the contrary, high AMT means a stable atmosphere jeopardizing the formation of precipitation. Once formed, however, the precipitation can

easily turn intensive and extreme, since the heated and humid air continuously provides energy and moisture.

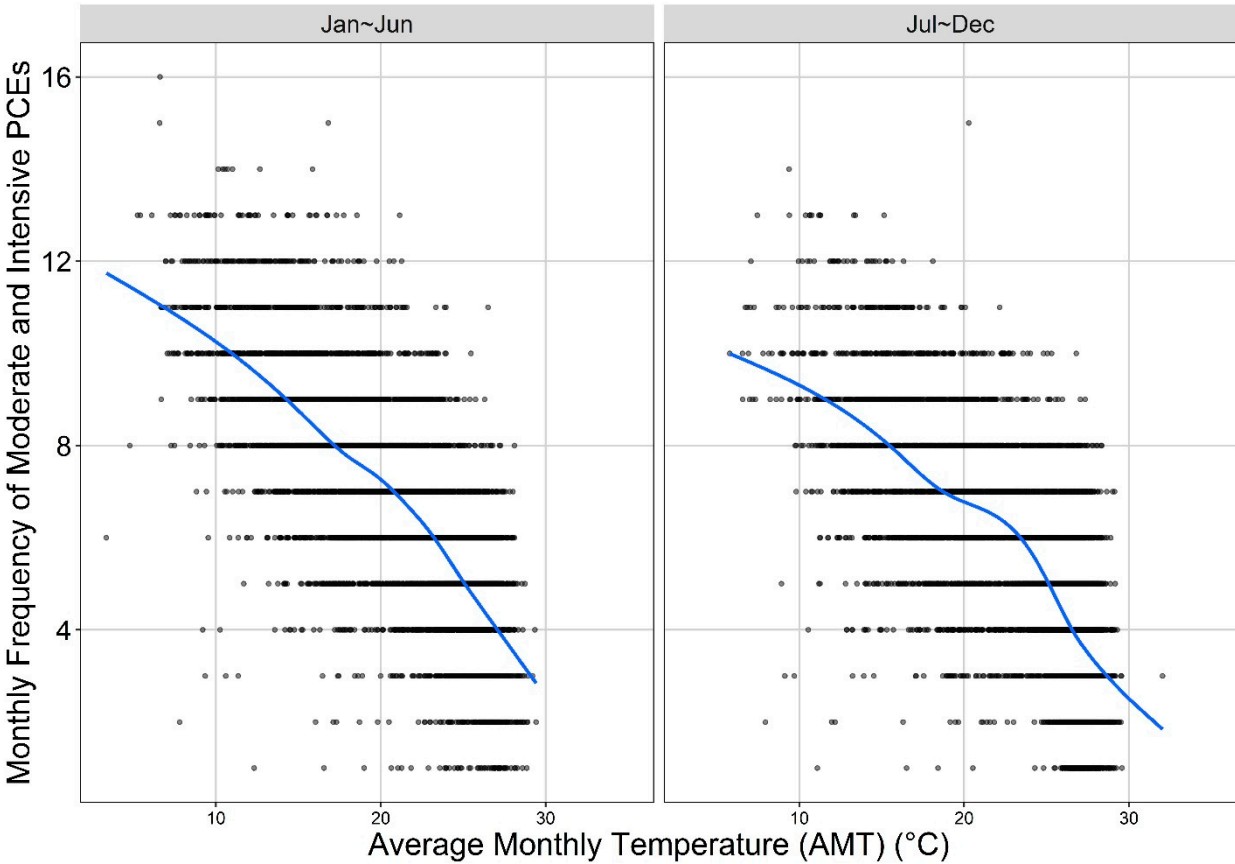

**Figure 4.** Association between monthly pressure change event (PCE) frequency and average monthly temperature (AMT) in local regression (blue lines).

The seasonal relationship between PD, AMT, and CEPC in different percentiles is investigated in Figure 5. Of the PD figures, 50%, 75%, and 95% are extracted from the pooled data for analysis. Other than the four-season partition, the wet season (June to October) and the dry season (November to May) are selected for discussion. Local regression lines are plotted for each season as well as the data for the whole year. Horizontally in Figure 5, the change in precipitation against AMT is subject to the magnitude and the type of PCE. Generally, PD changes more significantly under intensive CEPC in both directions. CEPC ranging from −300 hPa to 300 hPa does not provide sufficient power to extract moisture from the air when precipitation forms. For CEPC magnitudes beyond 300 hPa, either higher or lower, the change in PD is much more pronounced. Under intensive CEPCs, the C–C relationship is represented by the data when AMT is greater than 20 °C or when AMT is lower than 14 °C. For AMT > 20 °C, PD trend lines of all percentiles and both InPCE and DePCE lay parallel to the 7% rate. Discrepancies exist, however, when AMT < 14 °C. Specifically, a sub-C–C relationship can be observed for DePCE under CEPC < −300 hPa, and a super C–C relationship is shown for InPCE under CEPC > 300 hPa. This difference could be due to the mechanism of precipitation formation in different seasons, as discussed later. In Figure 5, PD shows various trends in different AMT and PCE types and percentiles. When AMT is higher than 20 °C, whether during the wet or dry season, PD increases at the 7% C–C rate for both intensive InPCE and intensive DePCE in all percentiles [9]. This is because intensive convections can completely extract moisture from the air to form precipitation. When AMT is lower than 12 °C, PD increases to higher than 7% under intensive InPCE conditions, while the rate for intensive DePCE is lower than 7%

in all percentiles. The main mechanism of precipitation formation is frontal movement, specifically cold fronts and warm fronts. The period of AMT < 12 °C corresponds to the spring season, during which the warm front dominates. A warm front is formed by the advance of a warm moist air mass and the simultaneous slow retreat of cold dry air. Typically, warm air moves from the southeast to the northwest in the northern hemisphere. Since warm air has a lower density, it rolls up and over the cold air and can cause light to moderate precipitation over a large geographic area. In this case, the ground-measured temperature is usually lower than the warm precipitation, causing air mass to roll up at high altitude. In other words, the change in PD of AMT < 12 °C actually corresponds to a wider temperature change than that observed from the ground.

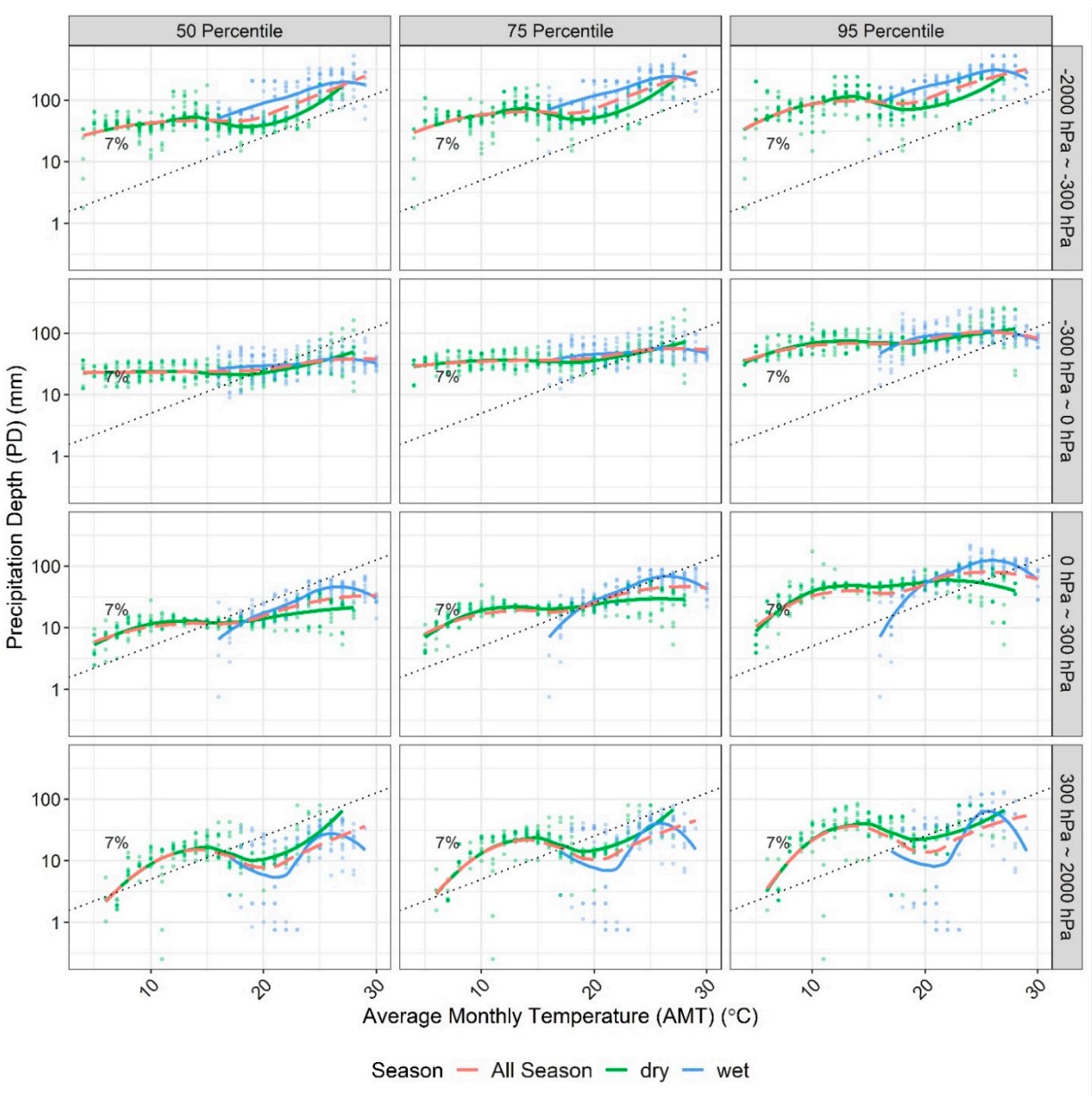

**Figure 5.** Seasonal relationships between precipitation depth (PD), average monthly temperature (AMT), and cumulative event pressure change (CEPC) in different percentiles.

Figure 6 investigates the associations between precipitation and AH under conditions of different temperatures (seasons) and CEPC extremes. The data used in Figure 6 are the

extreme PCEs (<−300 hPa or >300 hPa) that contain a 48 h window centered on the start of a precipitation event with two dry periods precedent and after the precipitation event. Such an isolated precipitation event can help reflect the AH response to the precipitation formation under extreme weather conditions and different seasons. In general, AH has an obvious dipping and recovering pattern in the wet season, while the magnitude for CEPC < −300 hPa is slightly larger than CEPC > 300 hPa. In the dry season, the AH recovery after precipitation is more obvious for CEPC > 300 hPa. Both extreme conditions show a general decreasing trend of AH after precipitation, which might be caused by cold front invasions bringing a temperature drop. All conditions have a similar average RH, which indicates that the humid supply in Florida does not have an obvious seasonality and does not affect the analysis of AH pattern change. In addition, Figure 6 states that AH under extreme CEPC responds differently to seasons (wet and dry) and PCE types when precipitation presents. In the wet season, AH decreases before 6 h of precipitation and over-recovers after precipitation starts. In the dry season, however, neither the decrease nor the recovery is as pronounced as in the wet season. Given that RH in all conditions shows little difference, this change in AH response is mainly due to the difference in temperature between the two seasons. As per Figure 5, PD is always higher in the wet season than in the dry season under extreme DePCEs, which is indicated by the difference in the AH magnitude change in the upper two plots in Figure 6. On the other hand, PD for extreme InPCEs is on the same level for both seasons (see the plots on the bottom row in Figure 5), which can also be inferred by the similar magnitude of AH change shown in the bottom row of Figure 6.

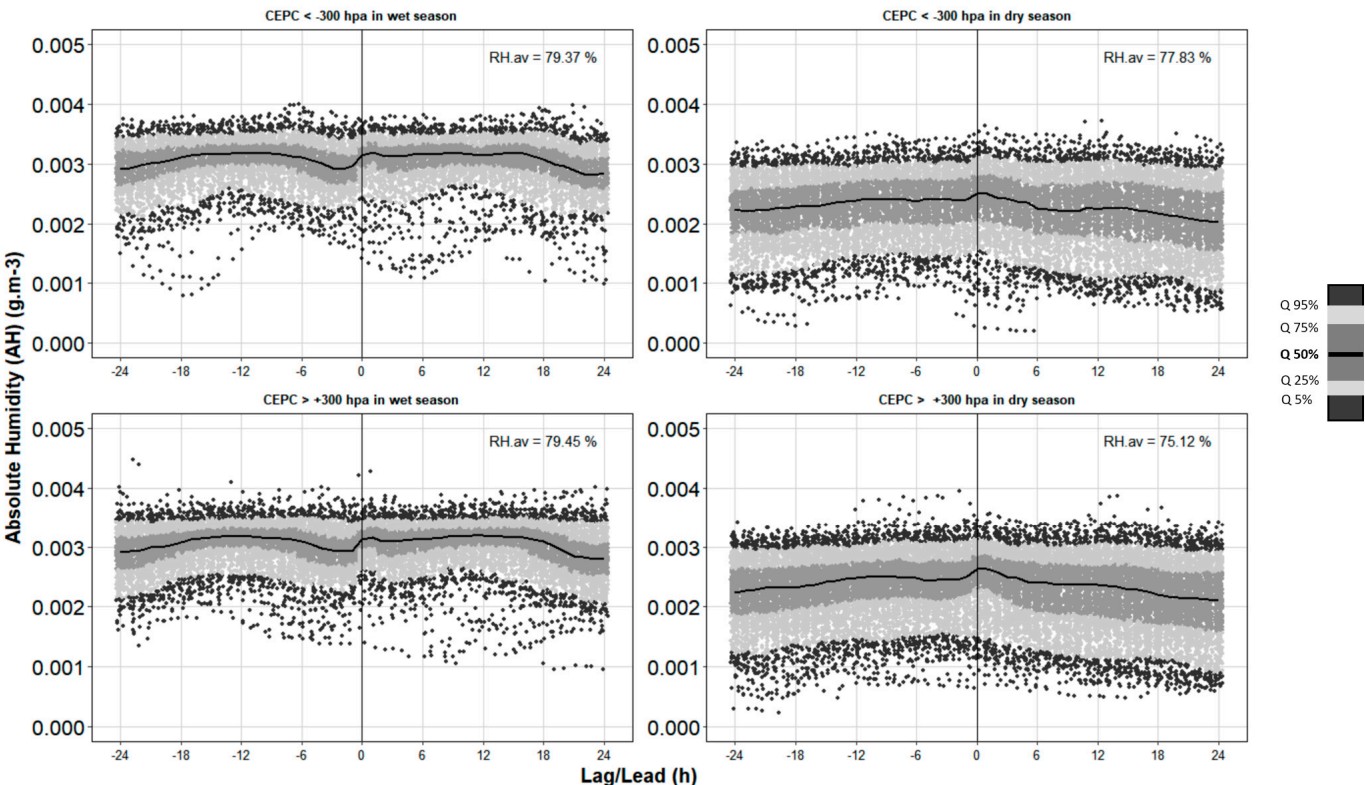

**Figure 6.** Quantile ranges of absolute humidity (AH) in extreme pressure change events (PCEs) responding to precipitation. The solid center line at 0 h indicates the start of precipitation. Seasons (dry and wet) and cumulative event pressure change (CEPC) are used to categorize the analysis [28].

## 5. Summary

Generally, this study aims to justify the physical concepts of how weather events are formed, using evidence from pressure change event (PCE)-based weather data analysis,

especially for extremes. Data from the Florida Automated Weather Network (FAWN) and National Centers for Environmental Information from NOAA (NCEI) in Florida are employed to explore the relationship between the formation mechanism of precipitation and PCE, with assistance from other types of measurements (e.g., temperature, relative humidity). Particularly, the results of this study respond to the following questions: (1) In different types of PCE process, how do precipitation depth (PD) and probability of precipitation (POP) change with cumulative event pressure change (CEPC)? The weather data collected by point source weather stations may have a low chance of observing all precipitation events, especially for increasing pressure change events (InPCE). Therefore, given the extremes are in a small sample size, such underestimation of actual POP and PD could be amplified. (2) For different temperatures, could precipitation represent Clausius–Clapeyron (C–C) under PCE conditions and humidity change patterns?

The following points present our conclusions:

1. The POP and PD are sensitive to the type and magnitude of PCE. For decreasing pressure change events (DePCE), PD and POP are generally positive to the magnitude of CEPC, while for InPCE, CEPC amplification corresponds to a decrease in PD and a stable POP.
2. The formation of precipitation is more likely to occur when the average monthly temperature (AMT) is low and PCE alternates frequently. During high AMTs, once the formation of precipitation occurs, precipitation can easily turn intensive and extreme.
3. PD trends at a similar level to the C–C rate (7%) for both intensive InPCEs and DePCEs when AMT is higher than 20 °C. This phenomenon is due to the intensive convections that extract most of the moisture from the air to form precipitation, when more moisture can be held by the air.
4. When AMT is lower than 12 °C, PD increases at a rate much higher than 7% in intensive InPCEs. Warm front movement in the spring could be the main cause of this phenomenon. Lifted warm air mass may carry more moisture than the capacity corresponding to the ground temperature observations, which means that the C–C relationship is likely obeyed at high altitude, but that a super C–C rate is observed on the ground.
5. Actual humidity responds differently to precipitation in different seasons and CEPC. Generally, more humidity is extracted during the wet season, while its magnitude for intensive DePCEs is higher than for intensive InPCEs.

The novelty and the unique findings of this work provide a potential method to explain the mechanism of precipitation, by refining the fundamental associations between precipitation and other weather parameters to PCE. As such, the results of this study may be of great value to climate researchers in furthering their understanding of climate change impacts and enhancing the existing methods of future weather projections.

**Author Contributions:** Conceptualization, Z.Y., C.Z., S.W., T.L. and J.B.; methodology, Z.Y., C.Z. and S.W.; software, C.Z. and S.W.; validation, Z.Y., C.Z. and S.W.; formal analysis, C.Z. and S.W.; investigation, C.Z. and S.W.; resources, C.Z. and S.W.; data curation, C.Z. and S.W.; writing—original draft preparation, C.Z. and S.W.; writing—review and editing, Z.Y.; visualization, C.Z. and S.W.; supervision, Z.Y. and J.B.; project administration, Z.Y.; funding acquisition, Z.Y., C.Z., S.W., T.L. and J.B. All authors have read and agreed to the published version of the manuscript.

**Funding:** This work is sponsored by the USDA/NRCS CIG grant (NR213A750013G018) awarded in 2021.

**Institutional Review Board Statement:** Not applicable.

**Informed Consent Statement:** Not applicable.

**Data Availability Statement:** The data used in this study are downloaded from NCEI and FAWN. All publicly available. For NCEI, the stations with the corresponding ID and data periods are (Pensacola (ID:13,899: 1948–2021), Tallahassee (ID:93,805: 1942–2021), Jacksonville (ID:13,889: 1948–2021), Orlando (ID:12,815: 1952–2021), Tampa (ID:12,842: 1940–2021) and Miami (ID:12,839: 1948–2021)).

FAWN's data is available at its FTP site (https://fawn.ifas.ufl.edu/data/fawnpub/, accessed on 21 January 2022).

**Conflicts of Interest:** The authors declare no conflict of interest.

## List of Acronyms

| Acronyms | Description |
| --- | --- |
| PCE | Pressure change event |
| CEPC | Cumulative event pressure change |
| DePCEs | Decreasing pressure change events |
| InPCEs | Increasing pressure change events |
| IntEDP | Inter-event dry period |
| POP | Probability of precipitation |
| PD | Precipitation depth |
| AMT | Average monthly temperature |
| C–C | Clausius–Clapeyron |
| NOAA | National Oceanic and Atmospheric Administration |
| NCEI | National Centers for Environmental Information from NOAA |
| FAWN | Florida Automated Weather Network |

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
