# Peer review of "Interpreting the Trends of Extreme Precipitation in Florida through Pressure Change"

_remotesensing, doi:10.3390/rs14061410_

Round 1

Reviewer 1 Report

I think the authors carried out a large amount of work to show the trends of extreme precipitation in Florida through pressure change. I think the authors need to present a detailed comprehensive methodology section. I recommend the manuscript for publication after the following major changes:

  • Your Current research focuses on the Pressure Change Events (PCE) sequence to understand extreme precipitation. Why don’t you consider K index, CAPE, or CIN parameters which had been generally accepted variables that are used in combination with a meteorologist’s experience to forecast severe precipitation events? The authors should explain this aspect in the introduction section. Otherwise, the readers cannot see the importance of your proposed methods over other studies.
  •  You should discuss climatic differences. could you please add the Köppen-Geiger climatic zones for the study area?
  • In the Study area section, you provided lots of quantitative information without any source or citation. Could you please provide related sources? You used multiple datasets with different sources with different temporal resolutions. How do you merge all those datasets into a common time series, please explain? It is necessary to report how the matching is carried out. This can have a significant impact on the results. I suggest adding a table that will include information about the resolution(temporal/spatial), sources, data ranges, etc.
  • In the discussion section, you should discuss your results vs previous research.
  • Can you provide a high impactful schematic diagram to understand your proposed methodology where the big impact of the results can be presented?
  • Your error analysis results are incomplete. Can you provide a table for error analysis in terms of systematic error (absolute mean relative error) and random error (normalized root mean square error) for the evaluation results? You can follow:

Khan et al. 2021:  Artificial Intelligence-Based Techniques for Rainfall Estimation Integrating Multisource Precipitation Datasets. Atmosphere 2021, 12, 1239.

Mei, et al. 2016: Evaluating satellite precipitation error propagation in runoff simulations of mountainous basins. J. Hydrometeor., 17, 1407–1423, https://doi.org/10.1175/JHM-D-15-0081.1

Author Response

COMMENTS FROM Reviewer #1:

Major comments:

  1. Your Current research focuses on the Pressure Change Events (PCE) sequence to understand extreme precipitation. Why don’t you consider K index, CAPE, or CIN parameters which had been generally accepted variables that are used in combination with a meteorologist’s experience to forecast severe precipitation events? The authors should explain this aspect in the introduction section. Otherwise, the readers cannot see the importance of your proposed methods over other studies.
    Meteorological parameters, such as K index, CAPE, and CIN, are all important in measuring the stability of atmospheric system. But agriculture or hydrology projects usually don’t have the required data for calculating these parameters. The motivation of our work is to model extremes from ground observation without considering geospatial dependence between different stations. We only have point-sourced data, that cannot directly calculate these parameters. In addition, the purpose of this study is to justify the relationship between precipitation and weather parameters using data in time series. There is no forecast or simulation of precipitation or extremes in our work.
  2.  You should discuss climatic differences. could you please add the Köppen-Geiger climatic zones for the study area?

Figure 1 has been updated, please refer to Line 132.

  1. In the Study area section, you provided lots of quantitative information without any source or citation. Could you please provide related sources? You used multiple datasets with different sources with different temporal resolutions. How do you merge all those datasets into a common time series, please explain? It is necessary to report how the matching is carried out. This can have a significant impact on the results. I suggest adding a table that will include information about the resolution(temporal/spatial), sources, data ranges, etc

We are assuming the “Study area section” you meant is the “Data” section in the manuscript. However, it is not clear about the “lots of quantitative information”. The sources and station information has been described in detail with citation provided. The first paragraph of the “Data preparation” section states the data quality regarding to missing points and how it would be excluded from the analysis. The second paragraph of the “Data preparation” section describes the methods of filling gaps with citations provided. The third paragraph of the “Data preparation” section provides the definition of PCE (with citation) and extreme events (with statistic justifications).

We agree with the importance of reporting “how the matching is carried out” from datasets from different sources. Actually, the data merge process has already been provided in the sentence of “While data from NCEI is hourly, FAWN’s data is in 15 min and aggregated into hourly by summing up precipitation and averaging other measurements.” in the first paragraph of the “Data preparation” section.

Given the study doesn’t consider the spatial resolution and only has two temporal resolutions, we think it is not necessary to tabulate them. 

  1. In the discussion section, you should discuss your results vs previous research.

Thanks for your advice. In fact, interpreting precipitation using PCE is pretty novel in hydrology which usually uses statistical assumptions for such investigations. The method of using PCE as the basis was recently published by Yu et al. (2018).  For Physical nexus, the classic meteorologic studies were done in the early 1900s which have been cited to support the scientific evidence. In addition, this study does not provide a new model that competes with previous research, it is an analysis of the observations with a new understanding of precipitation in hydrology.

  1. Can you provide a high impactful schematic diagram to understand your proposed methodology where the big impact of the results can be presented?

The second paragraph in the “Introduction” section describes the impact of this study. Most of the precipitation analyses in agriculture and hydrology are statistics based on whose results or parameters are very sensitive to the time and locations to which their data is attributed. That means a re-analysis is needed when new data is provided. This study investigates the physical evidence from the data that should hold everywhere on Earth ruled by the same physics system. Thus, reanalysis can be saved and the results from this study are portable to a different location and time.  A more detailed description can be found in the paper by Yu et al. 2018.

Yu, Z.; Miller, S.; Montalto, F.; Lall, U. The bridge between precipitation and temperature–Pressure Change Events: Modeling future non-stationary precipitation. Journal of Hydrology 2018, 562, 346-357.

  1. Your error analysis results are incomplete. Can you provide a table for error analysis in terms of systematic error (absolute mean relative error) and random error (normalized root mean square error) for the evaluation results? You can follow:

This is a verification study, and we did not establish any model for predicting precipitation. Figure 5 and Figure 6 are the findings derived from observations with no modeled results. With that given, error analysis for modeling is not applicable in this study since the findings derived from observations themselves are the facts that justify the physical concepts of precipitation. 

Reviewer 2 Report

Comments:

This manuscript entitled “Interpreting the trends of extreme precipitation in Florida 2 through pressure change” (remotesensing-1587908) was trying to justify the physical nexus of weather formation by the evidence from analyzing various weather observations on PCE basis, with emphasis on extreme precipitation. In my opinion, the structure of this manuscript is well organized and in an appropriate logical way. Therefore, I think this study has a significant value and an important practical meaning. However, the authors need to pay attention to some concerns of mine. Overall, I recommend this manuscript to be published in Remote Sensing after a minor revision. The specific comments can be seen as follows:

  1. Page 1 Line 37-39: Please rewrite this sentence.
  2. Page 4 Line 142: Delete the Indent before P’(t).
  3. Page 5 Line 157: “Where” should be “where” and no indent.

Author Response

COMMENTS FROM Reviewer #2:

Minor comments:

  1. Page 1 Line 37-39: Please rewrite this sentence.

Corrected Line 37-39

  1. Page 4 Line 142: Delete the Indent before P’(t).

Corrected Line 144-145

  1. Page 5 Line 157: “Where” should be “where” and no indent.

Corrected Line 160

Reviewer 3 Report

Interpreting the trends of extreme precipitation in Florida through pressure change

By Zhang et al.

This manuscript discusses the variation of the depth and probability of precipitation with cumulative event pressure change (CEPC) and average monthly temperatures. The authors present the responses of absolute humidity to precipitation in different seasons during 24 hours before and 24 hours after the start of the event. The analysis is performed over Florida. The introduction provides relevant references and the methodology sounds, but I do have some minor comments that should be easy to address:

  • what does “data technologies” means in the abstract
  • I suggest combining the result and discussion sections. I am saying this because it will be easier for the readers to find and understand the key findings of this investigation.
  • should have the 5 main elements of a map (lat, long, etc.).
  • Please be consistent in defining the abbreviations in the manuscript. please define them the first place they show up and do not define them every time you refer to them.
  • In the last part of the manuscript (section 6) please make specifically state your conclusions with some bullet points and please mention what is the novelty of your work and your unique findings.
  • Is it possible to report some statistics along with the visual illustrations and discussed relationships? It is better to judge with some metrics not just visual interpretations.

Author Response

COMMENTS FROM Reviewer #3:

Minor comments:

  1. what does “data technologies” means in the abstract

We found this term is too vague and doesn’t help in the abstract. So, we decided to delete the ‘data technologies’ in the sentence. Please refer to Line 6.

  1. I suggest combining the result and discussion sections. I am saying this because it will be easier for the readers to find and understand the key findings of this investigation.

Results and Discussions sections have been combined. Please refer to the line183-306

  1. should have the 5 main elements of a map (lat, long, etc.).

Figure 1 has been updated, please refer to Line 132.

  1. Please be consistent in defining the abbreviations in the manuscript. please define them the first place they show up and do not define them every time you refer to them.

Thanks for the advice, we double-checked the abbreviations. Because there are too many abbreviations (12) in this manuscript, which will make it hard for readers to clearly understand, we decide to spell out all those abbreviations in both figure captions and summary. So that reader who usually starts reading by figures and summary will not be confused.

  1. In the last part of the manuscript (section 6) please make specifically state your conclusions with some bullet points and please mention what is the novelty of your work and your unique findings.

Please check bullet points at Line 329-350.

For the novelty work and unique findings please refer to Line 351-353.

  1. Is it possible to report some statistics along with the visual illustrations and discussed relationships? It is better to judge with some metrics not just visual interpretations.

We wish that we can provide some statistics. However, this study is a qualitative analysis of the relationship between precipitation and humidity. We want to use this study to help us find the direction for our future work which is to quantify the relationship of precipitation subject to other weather parameters. Therefore, in future work, we will provide statistics to justify our justification.

Round 2

Reviewer 1 Report

The authors significantly improved the quality of the paper by addressing most of the previous comments. This research work will be very effective for the Water resources community. I recommend the manuscript for publication!